# Vitamin D insufficiency in COVID-19 and influenza A, and critical illness survivors: a cross-sectional study

Emma A Hurst,[1,2] Richard J Mellanby,[1] Ian Handel,[1] David M Griffith ,[3] Adriano G Rossi,[4] Timothy S Walsh,[4,5] Manu Shankar-Hari,[6,7] Jake Dunning,[8] Natalie Z Homer,[2] Scott G Denham,[2] Kerri Devine,[2] Paul A Holloway,[9] Shona C Moore,[10] Ryan S Thwaites,[11] Romit J Samanta,[12] Charlotte Summers ,[12] Hayley E Hardwick,[10] Wilna Oosthuyzen,[13] Lance Turtle,[10,14] Malcolm G Semple,[10,15] Peter J M Openshaw,[11] J Kenneth Baillie,[5,13] Clark D Russell ,[4,13] ISARIC4C Investigators

For numbered affiliations see end of article.

**Correspondence to**
Dr Clark D Russell;
clark.russell@ed.ac.uk

## ABSTRACT

**Objectives** The steroid hormone vitamin D has roles in immunomodulation and bone health. Insufficiency is associated with susceptibility to respiratory infections. We report 25-hydroxy vitamin D (25(OH)D) measurements in hospitalised people with COVID-19 and influenza A and in survivors of critical illness to test the hypotheses that vitamin D insufficiency scales with illness severity and persists in survivors.

**Design** Cross-sectional study.

**Setting and participants** Plasma was obtained from 295 hospitalised people with COVID-19 (International Severe Acute Respiratory and emerging Infections Consortium (ISARIC)/WHO Clinical Characterization Protocol for Severe Emerging Infections UK study), 93 with influenza A (Mechanisms of Severe Acute Influenza Consortium (MOSAIC) study, during the 2009–2010 H1N1 pandemic) and 139 survivors of non-selected critical illness (prior to the COVID-19 pandemic). Total 25(OH)D was measured by liquid chromatography-tandem mass spectrometry. Free 25(OH)D was measured by ELISA in COVID-19 samples.

**Outcome measures** Receipt of invasive mechanical ventilation (IMV) and in-hospital mortality.

**Results** Vitamin D insufficiency (total 25(OH)D 25–50 nmol/L) and deficiency (<25 nmol/L) were prevalent in COVID-19 (29.3% and 44.4%, respectively), influenza A (47.3% and 37.6%) and critical illness survivors (30.2% and 56.8%). In COVID-19 and influenza A, total 25(OH)D measured early in illness was lower in patients who received IMV (19.6 vs 31.9 nmol/L (p<0.0001) and 22.9 vs 31.1 nmol/L (p=0.0009), respectively). In COVID-19, biologically active free 25(OH)D correlated with total 25(OH)D and was lower in patients who received IMV, but was not associated with selected circulating inflammatory mediators.

**Conclusions** Vitamin D deficiency/insufficiency was present in majority of hospitalised patients with COVID-19 or influenza A and correlated with severity and persisted in critical illness survivors at concentrations expected to disrupt bone metabolism. These findings support early supplementation trials to determine if insufficiency is

## Strengths and limitations of this study

► Liquid chromatography-tandem mass spectrometry was used to quantify 25-hydroxy vitamin D (25(OH)D) in plasma samples from well-characterised hospitalised people with COVID-19 and influenza A and survivors of non-selected critical illness.

► Biologically active free 25(OH)D was measured by ELISA in COVID-19 plasma samples for the first time.

► Samples from people with COVID-19 and influenza A were obtained early in the course of the disease.

► Binary logistic regression multivariable models were used to assess the association of plasma 25(OH)D concentration with outcomes in COVID-19 and influenza A, correcting for other known relevant covariates.

► The observational nature of the study means it is not known whether vitamin D status led to poor clinical outcome or was a consequence of illness severity.

causal in progression to severe disease, and investigation of longer-term bone health outcomes.

## INTRODUCTION

Vitamin D metabolites contribute to bone metabolism, calcium homeostasis and immunomodulation. Vitamin D is a steroid pre-pro-hormone which is converted to the main circulating form 25-hydroxy vitamin D (25(OH)D), and subsequently to the active hormone 1,25 dihydroxy vitamin D. This second activation step occurs in the kidney, modulated by parathyroid hormone (PTH), for 'endocrine' calciotropic effects, and also under local control within extra-renal tissues, including immune cells, for direct action. These 'intracrine' actions on immune cells mediate antimicrobial and anti-inflammatory effects.[1] The majority of 25(OH)D circulates

bound to proteins, principally vitamin D binding protein (85%–90%), and the relatively small unbound ('free') fraction is available to immune cells.[2]

In the context of infectious diseases, vitamin D insufficiency (routinely determined by total 25(OH)D measurement) is associated with increased incidence and severity of respiratory tract infections,[3–5] including COVID-19.[6 7] A geographical association between vitamin D deficiency prevalence and COVID-19 incidence and mortality has been reported.[8] Free 25(OH)D has not yet been investigated in COVID-19, but this is required to fully understand vitamin D status during acute illness and any associations with systemic inflammation.[9] Clinical trials of vitamin D supplementation in respiratory diseases have returned mixed results.[10–13] Potential beneficial effects of vitamin D supplementation may be pathogen-specific and dependent on timing and route of administration. In addition to an interest in modifying acute illness outcomes, longer-term effects on bone health warrant consideration as critical illness is associated with loss of bone mineral density after recovery.[14]

In this cross-sectional study, we report measurements of total and free 25(OH)D in hospitalised people with COVID-19 and total 25(OH)D in hospitalised people with influenza A and survivors of critical illness. We use these three data sets to test the hypotheses that vitamin D insufficiency in severe respiratory virus infections scales with severity and persists in survivors of critical illness.

## METHODS
### Patients and sampling
#### COVID-19
The International Severe Acute Respiratory and emerging Infections Consortium (ISARIC) WHO Clinical Characterization Protocol for Severe Emerging Infections in the UK (CCP-UK) is an ongoing prospective cohort study of hospitalised people with COVID-19, which is recruiting in 308 hospitals in England, Scotland, Wales and Northern Ireland (National Institute for Health Research Clinical Research Network Central Portfolio Management System ID: 14152), delivered by the ISARIC Coronavirus Clinical Characterisation Consortium (ISARIC4C) investigators. The protocol, revision history, case report form and consent forms are available online at isaric4c.net. The ISARIC/WHO CCP-UK study was registered at https://www.isrctn.com/ISRCTN66726260 and designated an Urgent Public Health Research Study by the National Institute for Health Research UK. A prespecified case report form was used to collect data on patient characteristics, medical interventions received, and outcomes, as previously reported.[15]

#### Influenza A
Hospitalised patients with influenza A were recruited between 2009 and 2010 (the first and second H1N1 pandemic waves) and 2011 (the first postpandemic season) by the Mechanisms of Severe Acute Influenza

Consortium (MOSAIC) investigators, as previously reported.[16]

#### Non-selected critical illness survivors
We include a post-hoc analysis of the RECOVER (Evaluation of a Rehabilitation Complex Intervention for Patients Following Intensive Care Discharge) trial of intensive rehabilitation after critical illness.[17] Full eligibility criteria have been published previously; briefly, adults who had received invasive mechanical ventilation (IMV) for at least 48 hours and were considered well enough for discharge from the intensive care unit (ICU) were recruited. Patients gave additional consent for participation in a biomarker substudy and blood samples were collected at ICU discharge.[18]

All participants gave informed consent.

### Patient and public involvement
There was no patient or public involvement in this study.

### LC-MS/MS methods for total 25(OH)D analysis
EDTA plasma concentrations (on samples obtained on the day of enrolment to the study) of $25(OH)D_2$ and $25(OH)D_3$ isoforms were measured by liquid chromatography-tandem mass spectrometry (LC-MS/MS) and summed to derive the total 25(OH)D concentrations presented in the results. For patients with COVID-19 and critical illness survivors, analysis was performed by the Vitamin D Animal Laboratory using an assay which has been certified as proficient by the international Vitamin D External Quality Assessment Scheme and described in detail in an earlier manuscript, using 200 µL plasma.[19] The interassay precision (coefficient of variation, CV) of this method was <11.5% for both $25(OH)D_2$ and $25(OH)D_3$ analytes (online supplemental table 1). For patients with influenza A, analysis was performed using another LC-MS/MS method at a separate clinical biochemistry laboratory. The interassay precision of this method was <11% for $25(OH)D_2$ and <10% for $25(OH)D_3$ (online supplemental table 1). Full LC-MS/MS methods are presented in online supplemental table 1.

### Definition of vitamin D status
In addition to the absolute total 25(OH)D concentration, the relationship between vitamin D status and outcomes is often explored using a total 25(OH)D cutoff of 50 nmol/L to define populations that are vitamin D sufficient.[20] In this study, total 25(OH)D >50 nmol/L is reported as 'sufficient', 25–50 nmol/L as 'insufficient' and <25 nmol/L as 'deficient' (see online supplemental methods).

### Free 25(OH)D ELISA
Free 25(OH)D was measured using the Free 25OH Vitamin D ELISA (DIAsource ImmunoAssays SA, Belgium), following the manufacturer's instructions, using 10 µL of plasma. Absorbance was measured at 450 nm against a reference filter set at 630 nm using the Tecan Sunrise Microplate Reader (Tecan). GraphPad Prism (V.7.0e for

MacOS X) was used to perform a four-parameter logistic function to create the calibration curve in order to read the mean concentration of duplicate samples. The lower limit of detection (LLOD) of the assay was 2.4 pg/mL. The intra-assay repeatability (CV) was ≤5.5% across three concentrations (low, mid and high concentrations on the standard curve) and the interassay precision (CV) was <6.5% across the three concentrations, calculated based on Clinical and Laboratory Standards Institute EP05-A3 and reported in the manufacturer's guidelines. Two control samples (low and high concentrations) were analysed in each batch in duplicate and data were only reported for the batch if the results of the controls were within the acceptance range outlined on each control sample vial. Each calibrator, control and patient sample were assessed in duplicate and the results only reported if the CV of the replicates was <10%.

### Statistical analysis

For univariable analyses, the Shapiro-Wilk test was used to test for normal data distribution, and then appropriate tests, specified in the text, were used for comparisons. Associations between covariates and outcomes in COVID-19 and influenza A were assessed with binary logistic regression multivariable models. Sex, age, illness duration at time of sampling and comorbidity count were chosen as covariates. The comorbidity count was derived from the same comorbidities (table 1) from the two cohorts. To allow for potential non-linear relationship between predictors and the probability of an outcome, the models included smoothed thin plate regression spline terms for age, illness duration at time of sampling, comorbidity count and 25(OH)D concentrations. Multivariable models were estimated using the *gam*() function

| Table 1 | Characteristics of included patients | | | |
|---|---|---|---|---|
| **Characteristics** | **COVID-19 (n=259)** | **Influenza A (n=93)** | **Critical illness survivors (n=139)** | **P value*** |
| Demographics | | | | |
| Age at admission, years† | 63 (52–73) | 43 (29–50) | 63 (53–70) | <0.0001 |
| Male sex | 175 (67.6) | 47 (50.5) | 85 (61.2) | 0.01 |
| Day of illness at the time of sampling† | 10 (6–16) | 7 (4–11) | 11 (6–18)‡ | <0.001§ |
| Comorbidities | | | | |
| Diabetes mellitus | 66 (25.5) | 10 (10.8) | 23 (16.5) | 0.005 |
| Chronic cardiac disease | 57 (22.4) | 17 (18.3) | 15 (10.8) | 0.02 |
| Obesity, clinician-defined | 44 (18.7) | 23 (24.7) | 28 (20.1) | 0.3 |
| Asthma | 41 (16.1) | 33 (35.5) | 26 (18.7) | 0.0002 |
| Chronic lung disease, not asthma | 35 (13.8) | 12 (12.9) | 24 (17.3) | 0.5 |
| Chronic kidney disease | 25 (9.9) | 4 (4.3) | NA | 0.1 |
| Neoplasia | 14 (5.6) | 9 (9.7) | NA | 0.2 |
| Moderate or severe liver disease | 3 (1.2) | 4 (4.3) | NA | 0.08 |
| Illness severity | | | | |
| Admission to critical care | 106 (40.9) | 32 (34.4) | 139 (100) | 0.3§ |
| Invasive mechanical ventilation | 67 (25.9) | 29 (31.2) | 139 (100) | 0.3§ |
| In-hospital mortality | 52 (20.1) | 12 (12.9) | 4 (2.9)¶ | 0.2§ |
| Total plasma 25(OH)D | | | | |
| Median (IQR), nmol/L | 28.5 (17.1–51.9) | 28.1 (20.2–37.9) | 23.7 (15.3–34.9) | 0.01 |
| Status | | | | |
| Sufficient (>50 nmol/L) | 68 (26.3) | 14 (15.1) | 18 (12.9) | 0.0002 |
| Insufficient (25–50 nmol/L) | 76 (29.3) | 44 (47.3) | 42 (30.2) | |
| Deficient (<25 nmol/L) | 115 (44.4) | 35 (37.6) | 79 (56.8) | |

Data are number (%) unless otherwise stated.
*Kruskal-Wallis, Mann-Whitney or $\chi^2$ test as appropriate.
†Median (IQR).
‡Length of ICU stay.
§Comparing COVID-19 and influenza A.
¶Death after discharge from ICU.
ICU, intensive care unit; NA, not available; 25(OH)D, 25-hydroxy vitamin D.

of the R *mgcv* package using the default, thin plate regression smoothers.[21][22] The upper limit of smoother dimensionality was set to 9 for all variables, excluding the comorbidity count, where it was set to 7 as this variable was discrete with seven levels. Smoother parameters were estimated with restricted maximum likelihood. 25(OH)D concentrations were below the LLOD for 92 patients (free) and 2 patients (total) in the COVID-19 cohort. For the regression models, 25(OH)D values for these patients were imputed as the LLOD for the relevant analyte divided by the square root of 2.[23] As this is a commonly used but arbitrary method the regression analysis was repeated using 0 and the LLOD as imputed values to assess the sensitivity of the result to this assumption. Effects for categorical covariates are reported as OR; smoothed continuous covariates are reported graphically. Statistical analyses were conducted in R using the *mgcv*, *tidyverse* and *gratia* packages.

## RESULTS
### Patient characteristics
Samples were obtained from 259 people hospitalised due to COVID-19 and 93 people hospitalised due to influenza A. Samples were also obtained from 139 critical illness survivors (prior to the COVID-19 pandemic) at the time of ICU discharge. Patient characteristics, including sampling time after symptom onset, are shown in table 1. For patients with COVID-19, samples were obtained a median of 3 days (IQR 2–6) after hospital admission. Patients with influenza A were younger, more likely to be female and more likely to have asthma compared with the other cohorts. Receipt of IMV and in-hospital mortality did not differ between COVID-19 and influenza A. The WHO Ordinal Severity Scale scores for people with COVID-19 are shown in online supplemental figure 1, illustrating that the cohort is representative of the full spectrum of disease severity in hospitalised people. Details on ethnicity were available for the COVID-19 and influenza A cohorts. No differences in total 25(OH) D were observed between ethnic groups, but only small numbers of participants were from non-white groups (COVID-19 65 of 259, influenza A 25 of 93; online supplemental figure 2). All samples from people with influenza A were collected between the months of November and February (63.4% in December), whereas all samples from people with COVID-19 were collected between March and June (67.6% in April). However, the distribution of total 25(OH)D measurements did not differ when stratified by month (online supplemental figure 3). Total 25(OH)D concentration was lower in all three patient cohorts when compared with healthy controls (n=36; online supplemental figure 4), but the healthy control samples were obtained between the months of June and September.

### Total 25(OH)D correlates with severity in COVID-19
The majority of patients with COVID-19 had total 25(OH) D concentrations indicative of vitamin D insufficiency (29.3%) or deficiency (44.4%; table 1). Total 25(OH)D was lower in men than in women (median 26.8 (IQR 14.1–47.4) nmol/L vs median 31.7 (IQR 20.1–63.8) nmol/L, p=0.01) and weakly positively correlated with increased age (Pearson's r=0.25, p<0.0001).

When stratified by receipt of IMV as a marker of illness severity, total 25(OH)D differed significantly with a median concentration of 19.6 nmol/L (IQR 12.6–32.3) in patients receiving IMV compared with 31.9 nmol/L (IQR 20.0–58.3) in the remainder of the cohort (p<0.0001; figure 1A). When total 25(OH)D was stratified by associated vitamin D status, patients receiving IMV were more likely to be insufficient/deficient (figure 1A). Among patients who received IMV, 64.2% (43 of 67) were deficient and 26.9% (18 of 67) were insufficient. Total 25(OH)D concentration was also associated with in-hospital mortality (median 23.2 nmol/L (IQR 15.4–39.9) in non-survivors vs 29.5 nmol/L (IQR 17.2–55.4) in survivors, p=0.01). Total 25(OH)D concentrations were divided into quartiles and the proportion of patients who received IMV was compared (figure 1B). The lowest quartile (≤17.3 nmol/L) had the highest proportion of patients receiving IMV (43.1%). The middle quartiles were similar (26.6% and 25.0%), with the highest 25(OH) D quartile (>51.8 nmol/L) containing the lowest proportion receiving IMV (7.8%, $\chi^2$ p=0.0001).

Obesity is a risk factor for severity and mortality in COVID-19 and can be associated with vitamin D deficiency.[15] However, there was no difference in total 25(OH) D concentration between patients with/without clinician-defined obesity (online supplemental figure 5). Inflammatory mediator measurements had previously been performed on plasma samples from 66 patients included in this study.[24] Correlation matrix analysis demonstrated that total 25(OH)D was not significantly associated with circulating markers of systemic inflammation demonstrated to be involved in COVID-19 pathogenesis (online supplemental figure 6).

Multivariable analyses confirmed that total 25(OH) D concentration and vitamin D status (not sufficient) were both independently and negatively associated with receipt of IMV (table 2, figure 2A and online supplemental table 2). Two patients had total 25(OH)D concentrations below the LLOD; using 0 and LLOD, instead of LLOD divided by the square root of 2, had no substantive effect on significance of covariates or their effect sizes. Vitamin D status was also independently associated with in-hospital mortality, but total 25(OH)D concentration was not (online supplemental table 2 and online supplemental figure 7A).

### Total 25(OH)D correlates with severity in influenza A
We then extended these observations to total 25(OH) D concentrations measured in people hospitalised with influenza A. Total 25(OH)D was not associated with age (p=0.1) or sex (p=0.8). Similar to our findings in COVID-19, the majority of patients had total 25(OH)D concentrations indicative of vitamin D insufficiency (47.3%) or

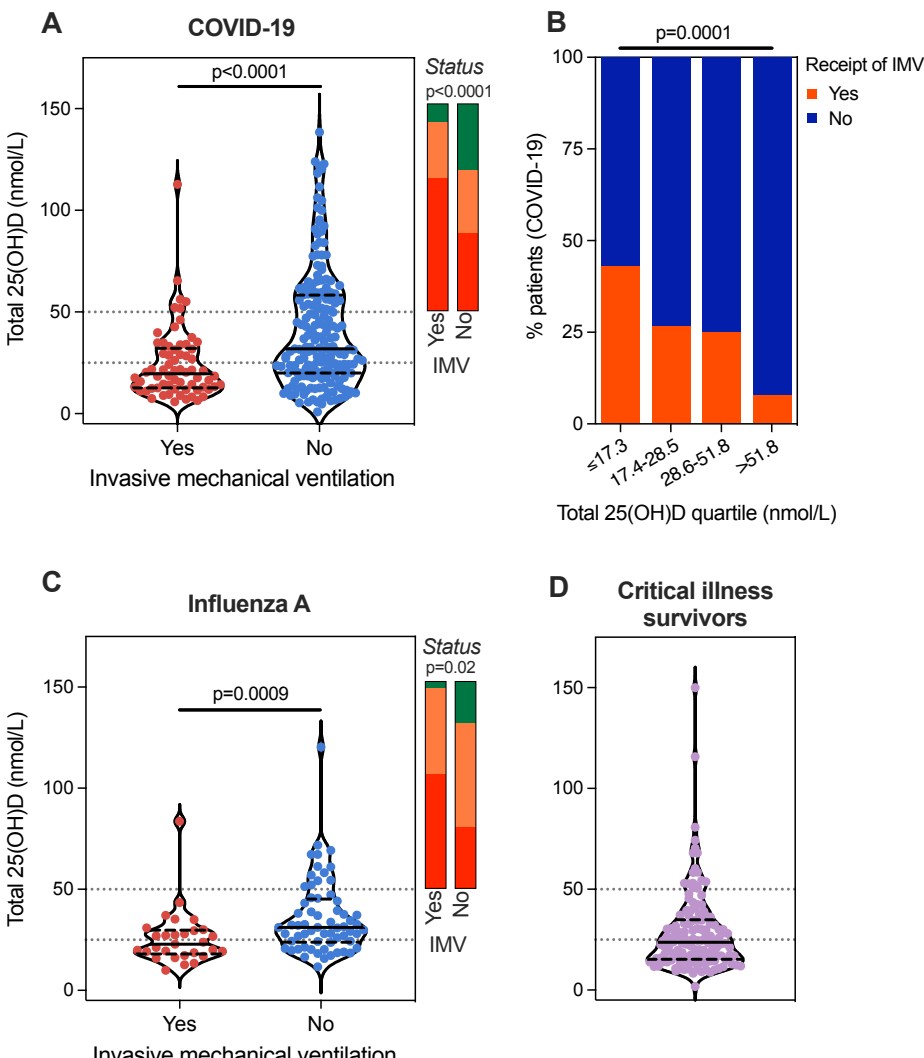

**Figure 1** Total 25(OH)D in COVID-19 and influenza A and in survivors of critical illness. (A) Total 25(OH)D concentrations in patients with COVID-19 (n=295) stratified by receipt of IMV. (B) Total 25(OH)D concentrations from patients with COVID-19 were divided into quartiles and the proportion of patients who received IMV in each quartile was compared by $\chi^2$ test. (C) Total 25(OH)D concentrations in patients with influenza A (from 2009 H1N1 pandemic, n=93) stratified by receipt of IMV. For (A) and (C), groups were compared by Mann-Whitney test. The stacked bar charts represent the proportion of patients in each subgroup with sufficient (green), insufficient (orange) or deficient (red) total vitamin D status, compared by $\chi^2$ test. (D) Total 25(OH)D concentrations in non-selected critical illness survivors (n=139, recruited prior to the COVID-19 pandemic) at the time of ICU discharge. On violin plots of total 25(OH)D concentrations (nmol/L), the solid line within the plot represents the median and the dashed lines represent the IQR. The dotted lines on the y-axis represent the thresholds for total vitamin D insufficiency (25–50 nmol/L) and deficiency (<25 nmol/L). 25(OH)D, 25-hydroxy vitamin D; ICU, intensive care unit; IMV, invasive mechanical ventilation.

deficiency (37.6%; table 1). When stratified by receipt of IMV, total 25(OH)D was lower in patients receiving IMV (median 22.9 nmol/L, IQR 18.0–29.8) compared with the remainder of the cohort (median 31.1 nmol/L, IQR 23.8–45.2, p=0.0009) and these patients were more likely to be vitamin D insufficient/deficient (figure 1C). Total 25(OH)D was lower in non-survivors compared with survivors (median 22.1 nmol/L (IQR 17.6–34.1) vs 29.2 nmol/L (IQR 20.6–38.5)), but this was not statistically significant (p=0.2). Multivariable analysis confirmed an independent negative association between total 25(OH)D and receipt of IMV but not in-hospital

mortality (figure 2B, table 2, online supplemental table 3 and online supplemental figure 7B).

### Vitamin D deficiency persists in survivors of critical illness
In survivors of non-selected critical illness, at the time of ICU discharge the median total 25(OH)D concentration was 22.9 nmol/L (IQR 14.6–34.6), similar to concentrations in patients with COVID-19/influenza A who required IMV or did not survive. The majority of patients had total 25(OH)D concentrations indicative of vitamin D deficiency (56.8%) or insufficiency (30.2%; figure 1D and table 1). Total 25(OH)D concentration was

Table 2 Multivariable analyses of 25(OH)D concentration and outcomes

| Variable | OR | P value |
|---|---|---|
| Total 25(OH)D | | |
| *COVID-19: receipt of IMV* | | |
| Male sex | 2.33 (1.13–4.78) | 0.022 |
| Comorbidity count* | – | 0.487 |
| Total 25(OH)D* | – | 0.001 |
| Day of illness* | – | 0.386 |
| Age* | – | 0.061 |
| *Influenza A: receipt of IMV* | | |
| Male sex | 2.22 (0.54–9.06) | 0.27 |
| Comorbidity count* | – | 0.15 |
| Total 25(OH)D* | – | 0.016 |
| Day of illness* | – | 0.001 |
| Age* | – | 0.19 |
| Free 25(OH)D | | |
| *COVID-19: receipt of IMV* | | |
| Male sex | 2.53 (1.24–5.314) | 0.011 |
| Comorbidity count* | – | 0.605 |
| Free 25(OH)D* | – | 0.006 |
| Day of illness* | – | 0.577 |
| Age* | – | 0.053 |
| *COVID-19: in-hospital mortality* | | |
| Male sex | 2.78 (1.25–6.17) | 0.012 |
| Comorbidity count* | – | 0.022 |
| Free 25(OH)D* | – | 0.025 |
| Day of illness* | – | 0.795 |
| Age* | – | 0.041 |

*Smoothed.
IMV, invasive mechanical ventilation; 25(OH)D, 25-hydroxy vitamin D.

not associated with age (p=0.7), sex (p=0.7) or length of ICU stay (p=0.8). Measurements were not available from earlier in these patients' illnesses.

### Free 25(OH)D correlates with severity in COVID-19

In patients with COVID-19, we found a strong correlation between free and total 25(OH)D concentrations (r=0.79, p<0.0001) (figure 3A). Free 25(OH)D was lower in patients receiving IMV (median 2.4 pg/mL (IQR 2.4–3.4) vs 3.6 pg/mL (IQR 2.4–5.7), p<0.0001; figure 3B) but was not statistically different between survivors and non-survivors on univariable analysis (median 2.8 pg/mL (IQR 2.4–4.4) vs 3.3 pg/mL (IQR 2.4–5.3), p=0.2). In multivariable analysis, free 25(OH)D was negatively associated with both receipt of IMV and in-hospital mortality (figure 2B–D and table 2). Free 25(OH)D was not associated with plasma inflammatory mediator concentrations (online supplemental figure 6).

## DISCUSSION

Vitamin D insufficiency was prevalent and scaled with severity in patients with COVID-19 and influenza A and insufficiency persisted in survivors of critical illness. As determined by total 25(OH)D measurement, 73% of patients with COVID-19, 84% of patients with influenza A and 87% of critical illness survivors were vitamin D insufficient/deficient. We demonstrate evidence of a strong association between vitamin D status (insufficiency/deficiency) during illness and both COVID-19 severity (receipt of IMV) and in-hospital mortality, with relevant confounders such as sex, age, comorbidities and day of illness adjusted for. This observation was replicated in influenza A, but the smaller sample size (n=93 compared with n=259) limited multivariable analyses. For the first time, we demonstrate a similar strong negative association between free 25(OH)D and COVID-19 disease severity and mortality. The results from this study extend earlier findings from other observational studies reporting associations between vitamin D status and SARS-CoV-2 infection and COVID-19 outcome.[6 7 25–27]

Vitamin D may beneficially modulate the host response against SARS-CoV-2 via intracrine immune signalling. Vitamin D enhances intracellular pathogen clearance, primarily via the induction of autophagy.[28] Importantly, the ability of macrophages to produce cathelicidin, which has antiviral activity against influenza virus and respiratory syncytial virus, correlates with circulating 25(OH)D concentrations.[29] Although antiviral effects of vitamin D have not yet been demonstrated in vitro for SARS-CoV-2, they have been demonstrated for other bacterial and viral pathogens.[30 31] Consistent with vitamin D having a role in local immunomodulation, neither free nor total 25(OH)D correlated with circulating markers of systemic inflammation involved in COVID-19 pathogenesis (including C-reactive protein, interleukin 6 and granulocyte-macrophage colony-stimulating factor[24]).

Evidence for the importance of free versus total 25(OH)D in relation to the mechanisms by which vitamin D exerts antimicrobial and anti-inflammatory functions has been demonstrated.[32 33] We now demonstrate that free 25(OH)D was negatively associated with COVID-19 severity and in-hospital mortality. Studies directly measuring free 25(OH)D and immune responses to infection or during critical illness are limited. In a study of 30 critically ill patients, supplementation with high-dose vitamin D increased free 25(OH)D and plasma cathelicidin concentrations.[34] Another study of 30 patients with sepsis reported similar results when they examined the effects of vitamin D supplementation on bioavailable (combined albumin-bound and free fraction) 25(OH)D and cathelicidin concentrations.[35] Together, these findings suggest that low concentrations of free 25(OH)D may reduce vitamin D-induced antimicrobial and anti-inflammatory response, compromising immune defences.

We found that 30.2% of patients surviving critical illness and requiring IMV (prior to the COVID-19 pandemic) were vitamin D insufficient and 56.8% were deficient.

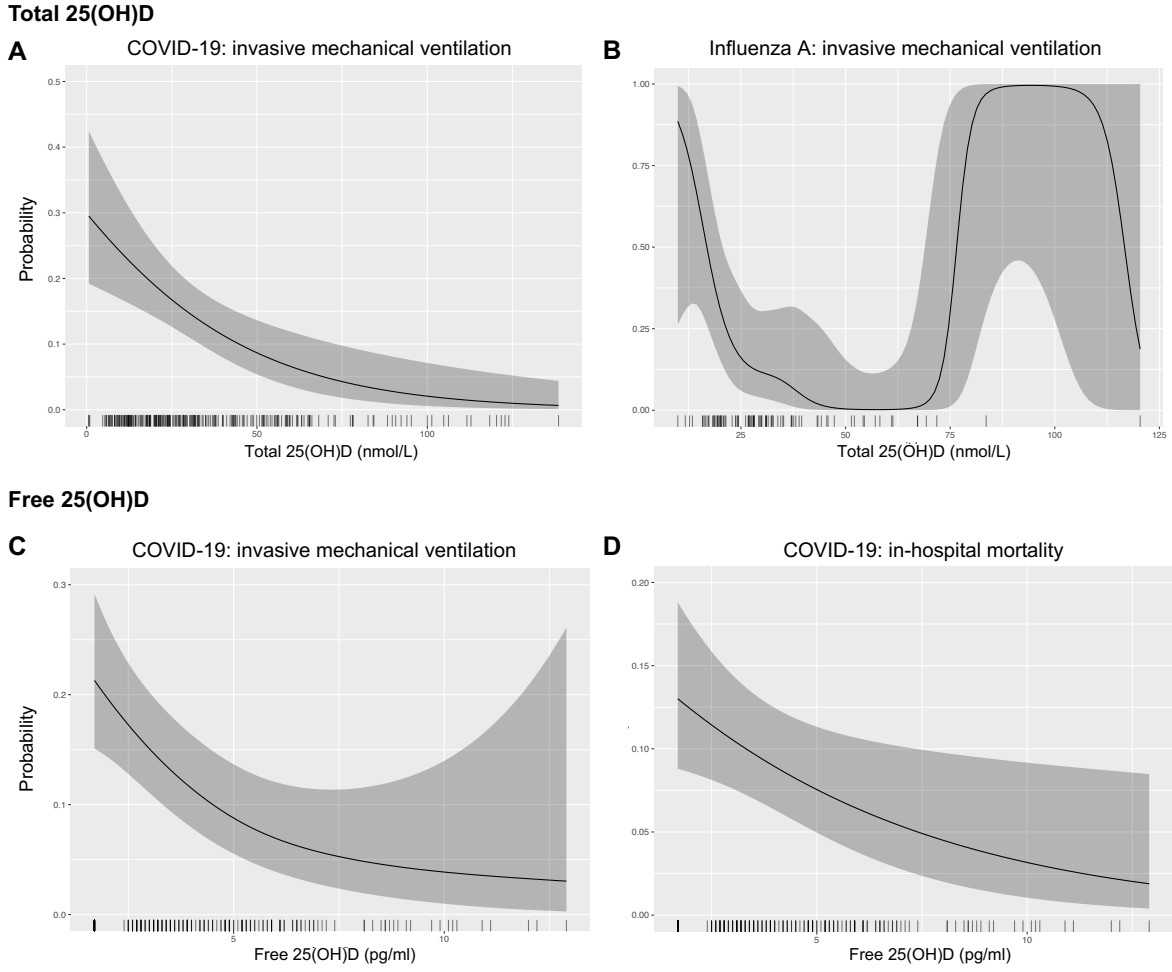

**Figure 2** Total and free 25(OH)D and outcomes in COVID-19 and influenza A. Smoothed predicted probability of outcomes (invasive mechanical ventilation (A. B and C) or in-hospital mortality (D)) versus total (A and B) or free (C and D) 25(OH)D concentration (with other covariates at mean values) from the binary logistic regression multivariable models. The grey ribbon represents the estimated 95% CI and the x-axis ticks show the observations. 25(OH)D, 25-hydroxy vitamin D.

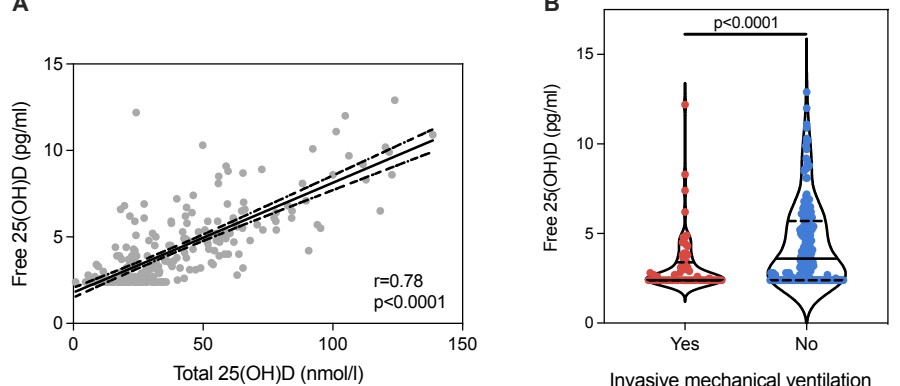

**Figure 3** Free 25(OH)D in COVID-19. (A) Simple linear regression line and 95% CI (dashed lines) representing the correlation between total and free 25(OH)D concentrations in COVID-19. (B) Violin plot of free 25(OH)D concentrations (pg/mL) in patients with COVID-19 stratified by receipt of invasive mechanical ventilation. The solid line within the plot represents the median and the dashed lines represent the IQR. Groups are compared by Mann-Whitney test. 25(OH)D, 25-hydroxy vitamin D.

Vitamin D deficiency is common in critical illness, with a reported prevalence of between 40% and 70% in observational studies of both adults and children worldwide.[36 37] Although some patients may enter ICU in a deficient state due to pre-existing disease and malnutrition, vitamin D metabolism is dysregulated in critical illness[38] and concentrations fall rapidly after ICU admission.[39] The mechanistic link between acute illness and vitamin D deficiency is likely to be multifactorial, including reduced dietary intake/absorption, reduced cutaneous synthesis due to lack of sunlight and wastage due to reductions in vitamin D binding protein.[40] Furthermore, vitamin D insufficiency/deficiency has been associated with a range of poor outcomes in critical illness.[37 41–43] Vitamin D insufficiency leads to secondary hyperparathyroidism and a concentration of 50 nmol/L total 25(OH)D is required for optimum PTH concentrations.[44] Of the critical illness survivors, 87% had total 25(OH)D <50 nmol/L, which would be associated with secondary hyperparathyroidism and the potential for associated loss of bone mineral density. Critical illness survivors suffer accelerated loss of bone mineral density in the year after ICU discharge (compared with matched controls) and increased 10-year fracture risk.[14] Our findings implicate vitamin D insufficiency in this process.

There is evidence that vitamin D supplementation can improve circulating total 25(OH)D concentrations in critically ill patients,[34 35 45] but evidence of a beneficial effect on outcomes is less clear. High-dose vitamin D supplementation in COVID-19[13] and critical illness[45] has been shown to increase plasma 25(OH)D concentrations 7 days post supplementation, but no significant reduction in the length of hospital stay or acute outcomes, including in-hospital mortality, admission to ICU or requirement for IMV, was demonstrated.[13 45 46] Longer-term outcomes such as bone health have not been evaluated. Conversely, two randomised trials reported high-dose 25(OH)D$_3$ (instead of vitamin D$_3$ as in the above-mentioned studies) on admission and then subsequent doses on either days 3 and 7, then weekly (n=76),[47] or days 3, 7, 15 and 30 (n=838),[48] were less likely to require ICU admission. We identified that vitamin D insufficiency was present early in the course of COVID-19 and influenza A (10 and 7 days after symptom onset, respectively), indicating that timing of supplementation may be an important factor when designing future supplementation studies. We propose that future studies examining effects on disease progression should investigate the effects of vitamin D supplementation given earlier in the course of the disease, closer to symptom onset rather than after hospitalisation. Evidence from an observational study of vitamin D supplementation usage supports this approach.[49] In a cohort of 8297 people with SARS-CoV-2 test results available, habitual vitamin D supplement intake prior to the pandemic was associated with a reduced risk of a positive test result after correction for known confounders including demographics and comorbidities. Furthermore, despite a decline in vitamin D following cardiothoracic surgery, postoperative outcomes (including organ dysfunction and mortality) are still associated with preoperative vitamin D status.[50] This suggests that supplementation prior to illness onset can still be expected to improve outcomes despite the fall in vitamin D concentration during acute illness. The longer-term effects of persistent vitamin D insufficiency/deficiency in survivors of critical illness also require further investigation especially in the context of bone health, which could be independently evaluated using sequential measurement of bone turnover markers and serum PTH.

The current study has several important limitations. The observational design prevents any conclusions about a causal role of vitamin D status in poor clinical outcome being drawn. We cannot exclude the alternative explanation that the differences in vitamin D status were a consequence of illness severity. Although blood samples were obtained from people with COVID-19 and influenza A as soon after hospital admission as feasible, even these early measurements will still be subject to acute illness-related changes in vitamin D homeostasis. Healthy control data are presented (online supplemental figure 4), but these samples were obtained between the months of June and September, whereas samples from people with COVID-19 and influenza A were obtained between November and June (although no intermonth variation was observed). Data on prehospital or in-hospital vitamin D supplementation were not available. Finally, longer-term follow-up samples to assess vitamin D status in survivors of COVID-19 will be informative.

In conclusion, vitamin D deficiency/insufficiency was present in the majority of hospitalised patients with COVID-19 or influenza A and scaled with severity, highlighting that reduced concentrations of vitamin D are common to these disease states and distinct patient cohorts. For the first time, free and total 25(OH)D were studied in COVID-19, demonstrating consistent results. It is not clear whether vitamin D status led to poor clinical outcome or was a consequence of illness severity. Randomised trials will be necessary to determine whether a causal relationship exists between vitamin D early in the course of the disease and development of critical illness. Since vitamin D deficiency/insufficiency persisted at concentrations expected to disrupt bone metabolism in critical illness survivors, investigation of longer-term bone health outcomes is also warranted.

**Author affiliations**
[1]The Roslin Institute and Royal (Dick) School of Veterinary Studies, University of Edinburgh, Edinburgh, UK
[2]Mass Spectrometry Core, Edinburgh Clinical Research Facility, Queen's Medical Research Institute, Edinburgh, UK
[3]Molecular, Genetic and Population Health Sciences, University of Edinburgh, Edinburgh, UK
[4]University of Edinburgh Centre for Inflammation Research, The Queen's Medical Research Institute, Edinburgh, UK
[5]Intensive Care Unit, Royal Infirmary of Edinburgh, Edinburgh, UK
[6]Intensive Care Unit, Guy's and St Thomas' Hospital NHS Foundation Trust, London, UK

[7]Peter Gorer Department of Immunobiology, School of Immunology & Microbial Sciences, Kings College London, London, UK

[8]Centre for Tropical Medicine and Global Health, University of Oxford, Oxford, UK

[9]Faculty of Medicine, Department of Metabolism, Digestion and Reproduction, Imperial College London, London, UK

[10]NIHR Health Protection Research Unit in Emerging and Zoonotic Infections, Institute of Infection, Veterinary and Ecological Sciences, Faculty of Health and Life Sciences, University of Liverpool, Liverpool, UK

[11]National Heart and Lung Institute, Imperial College London, London, UK

[12]Department of Medicine, University of Cambridge, Cambridge, UK

[13]Division of Genetics and Genomics, Roslin Institute, University of Edinburgh, Edinburgh, UK

[14]Tropical and Infectious Disease Unit, Liverpool University Hospitals NHS Foundation Trust, Liverpool, UK

[15]Respiratory Medicine, Alder Hey Children's Hospital, Liverpool, UK

**Acknowledgements** The ISARIC4C work uses data provided by patients and collected by the NHS as part of their care and support. We are extremely grateful to the 2648 front-line NHS clinical and research staff and volunteer medical students who collected these data in challenging circumstances; and the generosity of the participants and their families for their individual contributions in these difficult times. Thanks to Joanna Simpson and Patricia Lee, who were involved in sample management and free 25(OH)D data collection during the analytical process. Thanks to Catherine Duff and Sharon Hannah, who assisted in the health and safety aspects of the project, and to the Clinical Biochemistry Department at Imperial College Healthcare NHS Trust for analysing the samples from patients with influenza in 2011. We also acknowledge the support of Jeremy J Farrar (Wellcome Trust) and Nahoko Shindo (WHO).

**Collaborators** ISARIC4C Investigators: Consortium Lead Investigator: J Kenneth Baillie. Chief Investigator: Malcolm G Semple. Co-Lead Investigator: Peter JM Openshaw. ISARIC Clinical Coordinator: Gail Carson. Coinvestigator: Beatrice Alex, Benjamin Bach, Wendy S Barclay, Debby Bogaert, Meera Chand, Graham S Cooke, Annemarie B Docherty, Jake Dunning, Ana da Silva Filipe, Tom Fletcher, Christopher A Green, Ewen M Harrison, Julian A Hiscox, Antonia Ying Wai Ho, Peter W Horby, Samreen Ijaz, Saye Khoo, Paul Klenerman, Andrew Law, Wei Shen Lim, Alexander J Mentzer, Laura Merson, Alison M Meynert, Mahdad Noursadeghi, Shona C Moore, Massimo Palmarini, William A Paxton, Georgios Pollakis, Nicholas Price, Andrew Rambaut, David L Robertson, Clark D Russell, Vanessa Sancho-Shimizu, Janet T Scott, Thushan de Silva, Louise Sigfrid, Tom Solomon, Shiranee Sriskandan, David Stuart, Charlotte Summers, Richard S Tedder, Emma C Thomson, AA Roger Thompson, Ryan S Thwaites, Lance CW Turtle, Rishi K Gupta, Maria Zambon. Project Manager: Hayley Hardwick, Chloe Donohue, Ruth Lyons, Fiona Griffiths, Wilna Oosthuyzen. Data Analyst: Lisa Norman, Riinu Pius, Thomas M Drake, Cameron J Fairfield, Stephen R Knight, Kenneth A Mclean, Derek Murphy, Catherine A Shaw. Data and Information System Manager: Jo Dalton, Michelle Girvan, Egle Saviciute, Stephanie Roberts, Janet Harrison, Laura Marsh, Marie Connor, Sophie Halpin, Clare Jackson, Carrol Gamble. Data Integration and Presentation: Gary Leeming, Andrew Law, Murray Wham, Sara Clohisey, Ross Hendry, James Scott-Brown. Material Management: William Greenhalf, Victoria Shaw, Sara McDonald. Patient Engagement: Seán Keating. Outbreak Laboratory Staff and Volunteers: Katie A Ahmed, Jane A Armstrong, Milton Ashworth, Innocent G Asiimwe, Siddharth Bakshi, Samantha L Barlow, Laura Booth, Benjamin Brennan, Katie Bullock, Benjamin WA Catterall, Jordan J Clark, Emily A Clarke, Sarah Cole, Louise Cooper, Helen Cox, Christopher Davis, Oslem Dincarslan, Chris Dunn, Philip Dyer, Angela Elliott, Anthony Evans, Lorna Finch, Lewis WS Fisher, Terry Foster, Isabel Garcia-Dorival, William Greenhalf, Philip Gunning, Catherine Hartley, Rebecca L Jensen, Christopher B Jones, Trevor R Jones, Shadia Khandaker, Katharine King, Robyn T Kiy, Chrysa Koukorava, Annette Lake, Suzannah Lant, Diane Latawiec, Lara Lavelle-Langham, Daniella Lefteri, Lauren Lett, Lucia A Livoti, Maria Mancini, Sarah McDonald, Laurence McEvoy, John McLauchlan, Soeren Metelmann, Nahida S Miah, Joanna Middleton, Joyce Mitchell, Shona C Moore, Ellen G Murphy, Rebekah Penrice-Randal, Jack Pilgrim, Tessa Prince, Will Reynolds, P Matthew Ridley, Debby Sales, Victoria E Shaw, Rebecca K Shears, Benjamin Small, Krishanthi S Subramaniam, Agnieska Szemiel, Aislynn Taggart, Jolanta Tanianis-Hughes, Jordan Thomas, Erwan Trochu, Libby van Tonder, Eve Wilcock, J Eunice Zhang, Lisa Flaherty, Nicole Maziere, Emily Cass, Alejandra Doce Carracedo, Nicola Carlucci, Anthony

Holmes, Hannah Massey. Edinburgh Laboratory Staff and Volunteers: Lee Murphy, Nicola Wrobel, Sarah McCafferty, Kirstie Morrice, Alan MacLean. Local Principal Investigators: Kayode Adeniji, Daniel Agranoff, Ken Agwuh, Dhiraj Ail, Erin L Aldera, Ana Alegria, Brian Angus, Abdul Ashish, Dougal Atkinson, Shahedal Bari, Gavin Barlow, Stella Barnass, Nicholas Barrett, Christopher Bassford, Sneha Basude, David Baxter, Michael Beadsworth, Jolanta Bernatoniene, John Berridge, Nicola Best, Pieter Bothma, David Chadwick, Robin Brittain-Long, Naomi Bulteel, Tom Burden, Andrew Burtenshaw, Vikki Caruth, David Chadwick, Duncan Chambler, Nigel Chee, Jenny Child, Srikanth Chukkambotla, Tom Clark, Paul Collini, Catherine Cosgrove, Jason Cupitt, Maria-Teresa Cutino-Moguel, Paul Dark, Chris Dawson, Samir Dervisevic, Phil Donnison, Sam Douthwaite, Ingrid DuRand, Ahilanandan Dushianthan, Tristan Dyer, Cariad Evans, Chi Eziefula, Chrisopher Fegan, Adam Finn, Duncan Fullerton, Sanjeev Garg, Sanjeev Garg, Atul Garg, Effrossyni Gkrania-Klotsas, Jo Godden, Arthur Goldsmith, Clive Graham, Elaine Hardy, Stuart Hartshorn, Daniel Harvey, Peter Havalda, Daniel B Hawcutt, Maria Hobrok, Luke Hodgson, Anil Hormis, Michael Jacobs, Susan Jain, Paul Jennings, Agilan Kaliappan, Vidya Kasipandian, Stephen Kegg, Michael Kelsey, Jason Kendall, Caroline Kerrison, Ian Kerslake, Oliver Koch, Gouri Koduri, George Koshy, Shondipon Laha, Steven Laird, Susan Larkin, Tamas Leiner, Patrick Lillie, James Limb, Vanessa Linnett, Jeff Little, Mark Lyttle, Michael MacMahon, Emily MacNaughton, Ravish Mankregod, Huw Masson, Elijah Matovu, Katherine McCullough, Ruth McEwen, Manjula Meda, Gary Mills, Jane Minton, Mariyam Mirfenderesky, Kavya Mohandas, Quen Mok, James Moon, Elinoor Moore, Patrick Morgan, Craig Morris, Katherine Mortimore, Samuel Moses, Mbiye Mpenge, Rohinton Mulla, Michael Murphy, Megan Nagel, Thapas Nagarajan, Mark Nelson, Matthew K O'Shea, Igor Otahal, Marlies Ostermann, Mark Pais, Carlo Palmieri, Selva Panchatsharam, Danai Papakonstantinou, Hassan Paraiso, Brij Patel, Natalie Pattison, Justin Pepperell, Mark Peters, Mandeep Phull, Stefania Pintus, Jagtur Singh Pooni, Frank Post, David Price, Rachel Prout, Nikolas Rae, Henrik Reschreiter, Tim Reynolds, Neil Richardson, Mark Roberts, Devender Roberts, Alistair Rose, Guy Rousseau, Brendan Ryan, Taranprit Saluja, Aarti Shah, Prad Shanmuga, Anil Sharma, Anna Shawcross, Jeremy Sizer, Manu Shankar-Hari, Richard Smith, Catherine Snelson, Nick Spittle, Nikki Staines, Tom Stambach, Richard Stewart, Pradeep Subudhi, Tamas Szakmany, Kate Tatham, Jo Thomas, Chris Thompson, Robert Thompson, Ascanio Tridente, Darell Tupper-Carey, Mary Twagira, Andrew Ustianowski, Nick Vallotton, Lisa Vincent-Smith, Shico Visuvanathan, Alan Vuylsteke, Sam Waddy, Rachel Wake, Andrew Walden, Ingeborg Welters, Tony Whitehouse, Paul Whittaker, Ashley Whittington, Padmasayee Papineni, Meme Wijesinghe, Martin Williams, Lawrence Wilson, Sarah Cole, Stephen Winchester, Martin Wiselka, Adam Wolverson, Daniel G Wootton, Andrew Workman, Bryan Yates, Peter Young.

**Contributors** Conceptualisation: EAH, RJM, CDR, LT, PJMO, JKB and MGS. Data curation: EAH, DMG, JD and CDR. Formal analysis: EAH, IH and CDR. Funding acquisition: RJM, PJMO, JKB and MGS. Investigation: EAH, RJM, IH, DMG, AGR, TSW, MS-H, NZH, SGD, KD, PAH, RST, RJS, JD, CS, LT, PJMO, JKB, MGS and CDR. Project administration: SCM, HEH and WO. Data visualisation: IH and CDR. Supervision: RJM, MGS, PJMO, JKB and CDR. Writing - original draft: EAH and CDR. Writing - review and editing: EAH, RJM, IH, DMG, AGR, TSW, MS-H, NZH, SGD, KD, PAH, RST, RJS, JD, CS, LT, PJMO, JKB, MGS and CDR. Validation: EAH, RJM and CDR. Guarantor: CDR.

**Funding** This work was supported by grants from the National Institute for Health Research (NIHR) (award CO-CIN-01), the Medical Research Council (grant MC_PC_19059) and by the NIHR Health Protection Research Unit (HPRU) in Emerging and Zoonotic Infections at the University of Liverpool in partnership with Public Health England (PHE), in collaboration with Liverpool School of Tropical Medicine and the University of Oxford (award 200907), NIHR HPRU in Respiratory Infections at Imperial College London with PHE (award 200927), Wellcome Trust and the Department for International Development (215091/Z/18/Z), and the Bill and Melinda Gates Foundation (OPP1209135), and the Liverpool Experimental Cancer Medicine Centre (grant reference: C18616/A25153), NIHR Biomedical Research Centre at Imperial College London (IS-BRC-1215-20013), EU Platform for European Preparedness Against (Re-)emerging Epidemics (PREPARE) (FP7 Project 602525) and NIHR Clinical Research Network for providing infrastructure support for this research. LT is supported by a Wellcome Trust Fellowship (205228/Z/16/Z). The MOSAIC study was supported by the Wellcome Trust (087805/Z/08/Z) and the Medical Research Council HIC-Vac network (MR/R005982/1). PJMO is supported by an NIHR Senior Investigator Award (award 201385). RJS is funded by a EMINENT GSK-NIHR research training fellowship supported by the Cambridge NIHR Biomedical Research Centre. The RECOVER trial was funded by the Chief Scientist Office Scotland (CZH/4/53 (trial registration number ISRCTN09412438). MS-H is funded by the National Institute for Health Research Clinician Scientist Award (CS-2016-16-011). We acknowledge the financial support of NHS Research

Scotland (NRS) for the Mass Spectrometry Core, Edinburgh Clinical Research Facility.

**Disclaimer** The views expressed in this publication are those of the author(s) and not necessarily those of the NHS, the National Institute for Health Research or the Department of Health and Social Care, or other funders.

**Competing interests** RJM and EAH are part of VitDAL, which provides a 25(OH)D assay service on a not-for-profit basis.

**Patient consent for publication** Not required.

**Ethics approval** Ethical approval for the ISARIC/WHO CCP-UK study (COVID-19) was given by the South Central Oxford C Research Ethics Committee in England (13/SC/0149), the Scotland A Research Ethics Committee (20/SS/0028), and the WHO Ethics Review Committee (RPC571 and RPC572, 25 April 2013). Ethical approval for the MOSAIC study (influenza A) was given by the NHS National Research Ethics Service, Outer West London Research Ethics Committee (09/H0709/52, 09/MRE00/67).

**Provenance and peer review** Not commissioned; externally peer reviewed.

**Data availability statement** Data are available upon reasonable request. Data generated by the ISARIC4C consortium is available for collaborative analysis projects through an independent data and materials access committee at isaric4c. net/sample_access.

**ORCID iDs**
David M Griffith http://orcid.org/0000-0001-9500-241X
Charlotte Summers http://orcid.org/0000-0002-7269-2873
Clark D Russell http://orcid.org/0000-0002-9873-8243

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
