## [Reviewer comments · BMJ Open]

ARTICLE DETAILS

TITLE (PROVISIONAL)	Vitamin D insufficiency in COVID-19 and influenza A, and critical illness survivors: a cross-sectional study
AUTHORS	Hurst, Emma A.; Mellanby, Richard; Handel, Ian; Griffith, David; Rossi, Adriano; Walsh, Timothy; Shankar-Hari, Manu; Dunning, Jake; Homer, Natalie Z.; Denham, Scott G.; Devine, Kerri; Holloway, Paul A.; Moore, Shona C.; Thwaites, Ryan S.; Samanta, Romit J.; Summers, Charlotte; Hardwick, Hayley E.; Oosthuyzen, Wilna; Turtle, Lance; Semple, Malcolm G; Openshaw, PJ; Baille, J; Russell, Clark

VERSION 1 – REVIEW

REVIEWER	Hewison, M University of Birmingham, Institute of Metabolism and Systems Research
REVIEW RETURNED	26-Jul-2021

GENERAL COMMENTS	The manuscript by Hurst et al describes a well organised and presented study in which the authors have investigated the impact of vitamin D status (serum 25-hydroxyvitamin D, 25D) on various outcomes for patients hospitalized with Covid-19, influenza A or survivors of non-specific critical illness. The data are a cross-sectional study of the impact of serum 25D on critical care patients, similar to several previously reported studies. However, the current study includes several important advances: 1) the study compares Covid-19 effects of 25D with other related hospital illness; 2) the study includes 25D measurements from samples taken relatively early after diagnosis; 3) the study compares effects of serum 25D with 'free' serum 25D – with the latter possibly providing a better marker of the immune function of vitamin D. Overall this is an important new statement on the links between vitamin D status and Covid-19 disease progression in hospitalized patients. Specific comments 1) The authors need to include (In Materials and Methods) much greater detail about the assay performance for the Free 25D ELISA kit.2) Presentation of the serum 25D data as % of patients who are deficient/sufficient etc is OK but it would be helpful to have greater
--

	indication of actual median serum 25D levels (some of these values are listed in the text but may be missed from the headline observations. For example, the serum 25D values for non-specific illness survivors are no different to those from Covid-19 patients who required IMV or who died. 3) In a follow-up to the previous comment, can the authors provide some explanation for the relatively low serum 25D levels in critical illness survivors? Was this measured later and is therefore lower? Is there change in some other factor – for example, vitamin D binding protein (DBP)? DBP may show acute phase react changes in patients with critical illness. If this is the case then do survivors of critical illness have low total 25D but adequate ‘free’ 25D? 4) The serum 25D data presented in the study were produced using the gold-standard LC-MS/MS methodology, and this is well described in the manuscript. However, the data presented are an amalgamation of 25D3 and 25D2 values for each patient. Have the authors looked at 25D3 and 25D2 values separately? For most people the 25D3 value will be the predominant form but it would be interesting to purely investigate this one form of 25D. This could be Supplemental data but it would be interesting to see if 25D3 alone provides a stronger marker of vitamin D function. 5) There have been many studies that have explored the impact of vitamin D status on Covid-19 hospital outcomes. These cross-sectional analyses are limited by the same causality versus consequence confusion. Can the authors shed some further light on this? In Table 1 the authors report ‘Day of illness at time of sampling’ with a spread of approximately 6-18 days. Also, the median 25D value for in-hospital mortality is slightly higher than the serum 25D for patients receiving IMV, whereas one might have expected the opposite. Consequently, can the authors provide some sort of analysis of the potential change of serum 25D with length of care? One of the strengths of this study is the relatively high patient number, so it would be interesting to try and interrogate the issue of whether serum 25D levels decline with time after initial diagnosis of illness – are values higher in those patients who are sampled closer to the start of illness? 6) In a follow-up to the previous question, the authors should provide some discussion of how serum 25D levels may decline if low serum 25D levels are in fact a consequence of illness. What are the proposed underlying mechanisms? This is often proposed as an explanation for low serum vitamin D in patients receiving acute care but is rarely adequately defined. 7) Can the authors provide any data on whether or not patients in the study were regular users of vitamin D supplements?
--	--

REVIEWER	Bychinin, Mikhail Federal Scientific and Clinical Center of Specialized Types of Medical Care and Medical Technologies of the Federal Medical and Biological Agency of Russia, Intensive Care Unit
REVIEW RETURNED	29-Jul-2021

GENERAL COMMENTS	This is a relevant study that assesses vitamin D status in hospitalized people with COVID-19 and influenza A, and survivors of critical illness. The investigators revealed that vitamin D insufficiency is common among these patient's groups and scales with illness severity and persists in survivors. Authors report measurement of biologically active free 25(OH)D in addition to total in COVID-19. That is one of the first studies that investigated free 25(OH)D in COVID-19 patients. The results of this study enhance already existing data concerning the association between vitamin D status and severe outcome in patients hospitalized with COVID-19. The manuscript is clearly written and the ultimate conclusion justified by the results. However, I have several questions and comments that, if addressed, I believe would improve the utility of the manuscript:  1. Page 10 of 48, line 10. ICU abbreviation needs to be written in full first time 2. Page 20 of 48, line 30 SARS-CoV-2 abbreviation needs to be written in full first time. 3. The supplementary Figure 5 All lab abbreviations need to be written in full. 4. The severity of illness in hospitalized patients with influenza A and COVID-19 at the time of the 25(OH)D measurement is not clear. It is well known that a critical illness leads to a decrease in 25(OH)D. This is also indicated by the authors themselves. Could the authors supplement patient characterization with severity scales. 5. The key question about this study, and many before this, is whether low vitamin D contributes to Covid-19 infection and mortality or whether it is simply a marker of poor health. This is why it would be fascinating to provide some additional values for vitamin D in similar populations that were uninfected or not admitted to hospital.
--

REVIEWER	Boucher, Barbara Joan Queen Mary University of London, The Blizard Institute [Hon Professor]
REVIEW RETURNED	29-Jul-2021

GENERAL COMMENTS	Review-bmjopen.Hurst-et-al-VitD-Insuff-post-covid-InfA-critical-July-21 This report presents findings on the degrees of vitamin D insufficiency/deficiency in patients hospitalized with Covid-19 or Influenza-A that show these disorders to mark illness as judged by needing ventilation/mortality]; it also shows that low vitamin D status post-recovery from critical illness [before the Covid-19 pandemic] also reflected the severity of such illness. The findings
--

were reported as supporting the view that that low vitamin D status was associated with illness severity in each of these various situations, warranting studies of early supplementation in such patients.

General comments.

1. It is already known that pre-Covid-pandemic vitamin D status [the Kaufmann et al study that is mentioned in the MS] and that pre-pandemic use of vitamin D supplements, but not other types of supplements, [the Ma et al. study using UK Biobank data , which should be quoted] are each associated prospectively with reduced Covid-19 risks. It is also known that higher pre-operative serum 25(OH)D values are associated with reduced infection risks in hospitalised patients who have had major surgery despite the fact that major surgery, like major infections, reduces 25(OH)D values] [e.g., your ref 38 + Ney J et al. 2019] -might you add such a report? The data available from this report and from the prospective studies just mentioned should allow the authors to state that improvement of vitamin D status would be a measure likely to reduce the risks of covid-19 and Influenza-A illness in populations where vitamin D inadequacy continues to be common [e.g., as reported by Lips, Cashman et al. 2019].

2. Free circulating 25(OH)D was correlated with total serum 25(OH)D concentration but not with circulating inflammatory markers; however, since 25(OH)D concentration is a major determinant of intracellular calcitriol formation in target tissues, and especially in immunoregulatory cells, and this activation is quite differently regulated from renal activation [see the many relevant reports from Hewison M and his colleagues] any inverse correlations of total serum 25(OH)D, especially 25(OH)D3 values, with circulating inflammatory markers or with pro-inflammatory cytokines known to be suppressed by vitamin D [e.g IL-6] or with anti-inflammatory cytokines known to be upregulated by vitamin D [e.g. IL-10] would indeed have provided important information, see later.

3. Obesity and diabetes reduce hepatic 25-hydroxylation of vitamin D, [e.g., Bouillon and Bikle on the fall of vitamin D dogmas, 2019] so that the adjustments made may need further consideration.

4. The study assumes that existing definitions of deficiency and insufficiency cut-offs are all that is needed in the present analyses but since thresholds vary widely for different effects it would be useful to know whether any further health benefits might emerge using different cut-offs, [e.g as in the D2d trial where T2DM risk reduction in pre-diabetics only emerged with achieved intra- trial 25(OH)D values of 100 nmol/l or above; Dawson-Hughes B, et al. 2021] and if so, this would affect the data reported on Page 13, lines 31+

5. In addition, it would be valuable to state what 25(OH)D value appears to be needed to achieve maximal protection against Covid-19/Inf-A risks, either from the analyses or from inspection of the curves plotted. This level appears to be ~100 nmol/l from the Kaufman et al data [in the paper quoted] and, if the present study

	supports that finding or suggests some other value, this would be valuable information. Specific comments. [by page and line number] Page 7, line 37 and lines 46-50, it would be useful to state what stage of illness or admission day the study aimed to make 25(OH)D measurements in the methods for both Covid-19 and Influenza-A , even though these may not have been achievable since this varied a lot as shown in Table 1, [since illness reduces these values]. Page 8, line ~52/3; as in all studies based on 25(OH)D data the CVs of the assay should be stated since even using this gold-standard methodology they can be quite high, so that readers do not need to look this up in the supplementary material. Page 9, lines 24+, the CVs for this methodology should also be stated, especially since those concentrations are much lower than those of total 25(OH)D. Page 10, line 8, isn't a measurement of anything usually called a variable, even when smoothed?? Page 11, lines 42-44 refer to healthy controls having higher 25(OH)D values but was this control data obtained during the same months as the patients were studied; please clarify since if it was, this is useful, but if not it cannot be assumed to reflect the effects of illness on 25(OH)D values. Page 13, line 50+, readers should be told a bit more on these inflammatory markers such as which are D sensitive, and a reference given for their association with Covid-19 risks. Furthermore, correlations should be looked for using different ranges [strata] of D status since biological association with nutrients are always S shaped [e.g see Lappe & Heaney R]. It might be, therefore, that associations of interest only emerge with correlations across certain ranges of 25(OH)D values [raw data plots might suggest such associations]. One might expect benefit to be seen at higher levels of 25(OH)D in view of the D2d reanalysis study, but ANY such findings would be important to report and indeed, for mortality on Page 14, line 10, the same is relevant to mortality analyses and Supp Fig 6a rather suggests an optimal effect range that could be worth examining though this looks much less likely for Influenza-A from supp Fig 6b. Page 18, line 17, consider inserting 'during illness' after '(insufficiency/deficiency; line 50 might need modification if allowing for the S shaped associations [see above] reveals anything useful]. Page 19, line 53, you should mention that such increases in deficiency could well add to the problems of long-Covid here since that problem can cause diabetes as well as muscle weakness; Page 20, line 14+, there is very new data out from Barcelona in this area, using calcifediol treatment, so that this section could be updated?
--	---

REVIEWER	Papadimitriou, Dimitrios Athens Medical Centre, Pediatric-Adolescent Endocrinology and Diabetes
REVIEW RETURNED	04-Aug-2021

GENERAL COMMENTS	I congratulate the authors for this excellent study design and outstanding paper. The importance of PTH levels and secondary hyperparathyroidism to bone health has been generally subsided, while its importance for bone health especially in children and adolescents is invaluable [PMCID: PMC6551135]. The fact that this parameter was correctly dealt by this study is very important. Not to forget also, that despite the global consensus on vitamin D's importance in skeletal health [Global Consensus Recommendations on Prevention and Management of Nutritional Rickets. J Clin Endocrinol Metab. 2016;101:394–415], no public health authority – and not even the WHO - advised for even minimum vitamin D replacement during the extensive lockdowns even in high risk patients or with established osteopenia or osteoporosis; not even mentioning about “The Big Vitamin D mistake” [PMID: 28768407] and the implications to global health from this disastrous policy that has been continuously defended – even as we write these lines - with no admittance of the statistically proven mistake in the calculation of the RDA for vitamin D. A very important finding is shown in Figure 3A, which could be further emphasized at the conclusion. The findings of this study show that probably early high-dose supplementation to positive COVID-19 patients could be a safe and useful strategy [Rastogi A, Bhansali A, Khare N, et al. Short term, high-dose vitamin D supplementation for COVID-19 disease: a randomised, placebo-controlled, study (SHADE study) Postgraduate Medical Journal Published Online First: 12 November 2020. doi: 10.1136/postgradmedj-2020-139065, BMJ Journals]. Below are some minor suggestions: Introduction L37: Here it could be added that in a recent ecological study [PMID: 34079693] it was shown that a higher 25(OH)D concentration may protect from serious-critical illness and death from COVID-19 disease - even more in the elderly - but does not seem to prevent severe acute respiratory syndrome coronavirus 2 from spreading. Introduction L41-42: This (with proper rephrasing) could be added to support the authors' approach: measurement of binding proteins and free vitamin D metabolites may be essential to create a more realistic approximation of vitamin D status [PMID: 23075939] These references could be also useful: Bioavailable 25(OH)D is inversely associated with illness severity in critically ill patients associated with increased mortality and morbidity in the intensive care unit [Madden, K.; Feldman, H.A.; Chun, R.F.; Smith, E.M.; Sullivan, R.M.; Agan, A.A.; Keisling, S.M.; Panoskaltis-Mortari, A.; Randolph, A.G. Critically Ill Children Have Low Vitamin D-Binding Protein, Influencing Bioavailability of Vitamin D. Ann Am Thorac Soc 2015, 12, 1654-1661, doi:10.1513/AnnalsATS.201503-160OC.] Vitamin D repletion in critical illness with a more aggressive dosing is showing promising results [Amrein, K.; Papinutti, A.; Mathew, E.;
--

	Vila, G.; Parekh, D. Vitamin D and critical illness: what endocrinology can learn from intensive care and vice versa. Endocr Connect 2018, 7, R304-R315, doi:10.1530/EC-18-0184] A major limitation and important parameter to discuss wherever the authors judge as appropriate: even if we had the possibility to measure at every infected person the total Vitamin D at the time of diagnosis, still it is possible that in those with an active disease, Vitamin D would have been lower than before infection, as it is rapidly “consumed” during illness [Caccialanza, R.; Laviano, A.; Lobascio, F.; Montagna, E.; Bruno, R.; Ludovisi, S.; Corsico, A.G.; Di Sabatino, A.; Belliato, M.; Calvi, M., et al. Early nutritional supplementation in non-critically ill patients hospitalized for the 2019 novel coronavirus disease (COVID-19): Rationale and feasibility of a shared pragmatic protocol. Nutrition 2020, 74, 110835-110835, doi:10.1016/j.nut.2020.110835], indicating that after a high dose early short term supplementation to achieve levels of 25(OH)D3 > 100 nmol/L, a relatively high daily supplementation would be required during illness, before returning to safe long term supplementation that does not require medical supervision [PMID: 34079693].
--	---

VERSION 1 – AUTHOR RESPONSE

Reviewer 1 (Prof. M Hewison)

1.1 The authors need to include (In Materials and Methods) much greater detail about the assay performance for the Free 25D ELISA kit.

Details of the assay performance including lower limit of detection, intra- and inter-assay precision, and acceptance criteria on coefficient of variation between replicates has now been included in the Materials and Methods section.

1.2 Presentation of the serum 25D data as % of patients who are deficient/sufficient etc is OK but it would be helpful to have greater indication of actual median serum 25D levels (some of these values are listed in the text but may be missed from the headline observations. For example, the serum 25D values for non -specific illness survivors are no different to those from Covid-19 patients who required IMV or who died.

The median total 25(OH)D values (and IQR) have been added to Table 1 (“Characteristics of included patients”) as suggested.

1.3 In a follow-up to the previous comment, can the authors provide some explanation for the relatively low serum 25D levels in critical illness survivors? Was this measured later and is therefore lower? Is there change in some other factor – for example, vitamin D binding protein (DBP)? DBP may show acute phase react changes in patients with critical illness. If this is the case then do survivors of critical illness have low total 25D but adequate ‘free’ 25D?

Total 25(OH)D has been shown by others to be reduced during critical illness but free 25(OH)D and DBP have not been measured at an equivalent timepoint to address this interesting comment definitively. At the time of recovery from critical illness it would be expected that the acute phase response would be resolving (distinct from measurement at the beginning or early stages of critical illness) so we would suggest the persistent low total concentrations would not be entirely due to a situation of low DBP but adequate free 25(OH)D. Ultimately, additional studies are required which measure both free and total 25(OH)D longitudinally in critically ill people in the ICU.

1.4 The serum 25D data presented in the study were produced using the gold-standard LC-MS/MS methodology, and this is well described in the manuscript. However, the data presented are an amalgamation of 25D3 and 25D2 values for each patient. Have the authors looked at 25D3 and 25D2 values separately? For most people the 25D3 value will be the predominant form but it would be interesting to purely investigate this one form of 25D. This could be Supplemental data but it would be interesting to see if 25D3 alone provides a stronger marker of vitamin D function.

Thank you for suggesting this analysis. In the Covid-19 cohort, 25D3 accounted for an average of 94.8% of the total 25(OH)D. When 25D3 was stratified by in-hospital mortality and receipt of invasive mechanical ventilation (IMV), the results obtained were very similar to those obtained using total 25(OH)D:

- No statistical difference for in-hospital mortality (median 21.7 vs. 27.5, $p=0.18$)
- A statistically significant difference for receipt of IMV (median 18.3 vs. 31.3, $p<0.0001$)

1.5 There have been many studies that have explored the impact of vitamin D status on Covid-19 hospital outcomes. These cross-sectional analyses are limited by the same causality versus consequence confusion. Can the authors shed some further light on this? In Table 1

the authors report ‘Day of illness at time of sampling’ with a spread of approximately 6-18 days. Also, the median 25D value for in-hospital mortality is slightly higher than the serum 25D for patients receiving IMV, whereas one might have expected the opposite. Consequently, can the authors provide some sort of analysis of the potential change of serum 25D with length of care? One of the

strengths of this study is the relatively high patient number, so it would be interesting to try and interrogate the issue of whether serum 25D levels decline with time after initial diagnosis of illness – are values higher in those patients who are sampled closer to the start of illness?

Thank you for suggesting this. We have plotted total 25(OH)D against illness duration (number of days since onset of symptoms) with a linear regression line and 95% confidence interval (below). There does not appear to be a significant change over time. We have made a similar observation in the larger cytokine analysis (Thwaites et al, Science Immunology 2021) where there are no clear temporal trends in proximal signalling inflammatory mediators (e.g. IL-6 and GM-CSF) after hospitalisation. We think this adds to the rationale for investigating early (pre-hospital) supplementation.

1.6 In a follow-up to the previous question, the authors should provide some discussion of how serum 25D levels may decline if low serum 25D levels are in fact a consequence of illness. What are the proposed underlying mechanisms? This is often proposed as an explanation for low serum vitamin D in patients receiving acute care but is rarely adequately defined.

We have included reference to this important point in the revised discussion. Briefly, we suggest that if low 25(OH)D levels are a consequence of acute illness then the mechanism will be multifactorial, involving reduced dietary intake, reduced sunlight exposure limiting cutaneous synthesis, and wastage due to reductions in vitamin D binding protein.

1.7 Can the authors provide any data on whether or not patients in the study were regular users of vitamin D supplements?

This data was not collected. This limitation has been highlighted in the discussion.

Reviewer 2 (Dr. M Bychinin)

2.1 Page 10 of 48, line 10. ICU abbreviation needs to be written in full first time

We have corrected this on first usage.

2.2 Page 20 of 48, line 30 SARS-CoV-2 abbreviation needs to be written in full first time.

We have corrected this

2.3 The supplementary Figure 5 All lab abbreviations need to be written in full.

Thank you for pointing out this omission – we have added the full names.

2.4 The severity of illness in hospitalized patients with influenza A and COVID-19 at the time of the 25(OH)D measurement is not clear. It is well known that a critical illness leads to a decrease in 25(OH)D. This is also indicated by the authors themselves. Could the authors supplement patient characterization with severity scales.

In Table 1 we include the proportion of patients in these cohorts who required admission to critical care and who received invasive mechanical ventilation. We have now included an additional supplementary waffle plot (Supplementary Figure 1) with the WHO ordinal severity scale score for the Covid-19 cohort. This shows that the severity of illness in this cohort is representative of the spectrum of disease in hospitalised people.

2.5 The key question about this study, and many before this, is whether low vitamin D contributes to Covid -19 infection and mortality or whether it is simply a marker of poor health. This is why it would be fascinating to provide some additional values for vitamin D in similar populations that were uninfected or not admitted to hospital.

We agree entirely with this comment, and the limitation is inherent to observational studies in this area. Supplementary Figure 4 shows the total 25(OH)D concentrations from the three study cohorts compared to healthy controls, demonstrating the healthy controls have higher concentrations. However, we have no additional data to address this important point. We have emphasised this limitation in our discussion.

Reviewer 3 (Prof. B Boucher)

3.1 It is already known that pre-Covid-pandemic vitamin D status [the Kaufmann et al study that is mentioned in the MS] and that pre-pandemic use of vitamin D supplements, but not other types of supplements, [the Ma et al. study using UK Biobank data , which should be quoted] are each associated prospectively with reduced Covid-19 risks. It is also known that higher pre-operative serum 25(OH)D values are associated with reduced infection risks in hospitalised patients who have had major surgery despite the fact that major surgery, like major infections, reduces 25(OH)D values] [e.g., your ref 38 + Ney J et al. 2019] -might you add such a report? The data available from this report and from the prospective studies just mentioned should allow the authors to state that improvement of vitamin D status would be a measure likely to reduce the risks of covid-19 and Influenza-A illness in populations where vitamin D inadequacy continues to be common [e.g., as reported by Lips, Cashman et al. 2019].

Thank you for highlighting these important additional references which we have now included. We completely agree with the points made here and have amended our discussion accordingly, related to habitual vitamin D supplementation and the association between pre-operative vitamin D status and post-operative outcomes.

3.2 Free circulating 25(OH)D was correlated with total serum 25(OH)D concentration but not with circulating inflammatory markers; however, since 25(OH)D concentration is a major determinant of intracellular calcitriol formation in target tissues, and especially in immunoregulatory cells, and this activation is quite differently regulated from renal activation [see the many relevant reports from Hewison M and his colleagues] any inverse correlations of total serum 25(OH)D, especially 25(OH)D₃ values, with circulating inflammatory markers or with pro-inflammatory cytokines known to be suppressed by vitamin D [e.g IL-6] or with anti-inflammatory cytokines known to be upregulated by vitamin D [e.g. IL-10] would indeed have provided important information, see later.

The correlation analysis shown in Supplementary Figure 6 (copied below) included free 25(OH)D in addition to total. A weak negative correlation was observed with free 25(OH)D and peripheral blood neutrophil count, but not with any other inflammatory mediators.

3.3 Obesity and diabetes reduce hepatic 25-hydroxylation of vitamin D, [e.g., Bouillon and Bikle on the fall of vitamin D dogmas, 2019] so that the adjustments made may need further consideration.

We stratified total 25(OH)D concentrations by presence or absence of obesity in people with Covid-19 but found no difference in this cohort so did not make any adjustments (Supplementary Figure 5). Only three people in the Covid-19 cohort had chronic liver disease so we have not investigated this further.

3.4 The study assumes that existing definitions of deficiency and insufficiency cut-offs are all that is needed in the present analyses but since thresholds vary widely for different effects it would be useful to know whether any further health benefits might emerge using different cut-offs, [e.g as in the D2d trial where T2DM risk reduction in pre-diabetics only emerged with achieved intra-trial 25(OH)D values of 100 nmol/l or above; Dawson-Hughes B, et al. 2021] and if so, this would affect the data reported on Page 13, lines 31+

Thank you for this suggestion. In the Covid-19 cohort (the largest disease cohort in the study) 248/259 (95.8%) people had a total 25(OH)D concentration <100nmol/L. When stratified by invasive mechanical ventilation (IMV), 66/248 vs. 1/11 received IMV and this difference was not statistically significant (Fisher's exact test p-value 0.29). When stratified by in-hospital mortality, 50/248 vs. 2/11 died in hospital (p = 1.0). Therefore, due to the very low number of people in our cohort with a total 25(OH)D ≥100nmol/L, we are underpowered to address this specific question related to a threshold of 100nmol/L.

3.5 In addition, it would be valuable to state what 25(OH)D value appears to be needed to achieve maximal protection against Covid-19/Inf-A risks, either from the analyses or from inspection of the curves plotted. This level appears to be ~100 nmol/l from the Kaufman et al data [in the paper quoted] and, if the present study supports that finding or suggests some other value, this would be valuable information.

Thank you for this suggestion. We used the Covid-19 cohort for this analysis to maximise the number of measurements available, stratified total 25(OH)D into quartiles (≤ 17.3 , 17.4-28.5, 28.6-51.8, >51.8 nmol/L) and performed a contingency table analysis on the proportion of people in each quartile who required invasive mechanical ventilation. This identified a statistically significant difference, plotted below (now included as an additional panel in Figure

1) suggesting total 25(OH)D >51.8 nmol/L is most beneficial. This is now reported in the results section.

3.6 Page 7, line 37 and lines 46-50, it would be useful to state what stage of illness or admission day the study aimed to make 25(OH)D measurements in the methods for both Covid-19 and Influenza-A , even though these may not have been achievable since this varied a lot as shown in Table 1, [since illness reduces these values].

We have clarified in the methods section that these measurements were made on samples obtained on the first day of enrolment to the study. For Covid-19 we know that enrolment occurred a median of 3 days after admission to hospital (IQR 2-6) and this is included in the text of the results.

3.7 Page 8, line ~52/3; as in all studies based on 25(OH)D data the CVs of the assay should be stated since even using this gold-standard methodology they can be quite high, so that readers do not need to look this up in the supplementary material.

We have added these details to the Materials and Methods section as suggested.

3.8 Page 9, lines 24+, the CVs for this methodology should also be stated, especially since those concentrations are much lower than those of total 25(OH)D.

Further details of the assay performance including inter- and intra-assay precision (CV) and acceptable CV criteria for duplicate measurements has now been included here.

3.9 Page 10, line 8, isn't a measurement of anything usually called a variable, even when smoothed??

"Parameters" here refers to the parameters of the smoothing method.

3.10 Page 11, lines 42 -44 refer to healthy controls having higher 25(OH)D values but was this control data obtained during the same months as the patients were studied; please clarify since if it was, this is useful, but if not it cannot be assumed to reflect the effects of illness on 25(OH)D values.

The healthy controls were recruited between June-September. This was included in the supplementary figure legend but we have added it to the main text for additional clarity and also to the limitations in the discussion. We included the healthy control data as a supplementary and not main result for this reason.

3.11 Page 13, line 50+, readers should be told a bit more on these inflammatory markers such as which are D sensitive, and a reference given for their association with Covid-19 risks. Furthermore, correlations should be looked for using different ranges [strata] of D status since biological association with nutrients are always S shaped [e.g see Lappe & Heaney R]. It might be, therefore, that associations of interest only emerge with correlations across certain ranges of 25(OH)D values [raw data plots might suggest such associations]. One might expect benefit to be seen at higher levels of 25(OH)D in view of the D2d reanalysis study, but ANY such findings would be important to report and indeed, for mortality on Page 14, line 10, the same is relevant to mortality analyses and Supp Fig 6a rather suggests an optimal effect range that could be worth examining though this looks much less likely for Influenza-A from supp Fig 6b.

Thank you for these informative comments. The reference for the cytokine data is Thwaites et al, Science Immunology 2021 and is cited in the text. Unfortunately, there was a limited overlap between the plasma samples that underwent multiplex cytokine analysis (for the Thwaites et al study) and vitamin D measurement. Only 66 patients had overlapping data. In addition to calculating Spearman correlations for each we have visualised the raw data and can see no evidence of an association. We discuss this and suggest it reflects a disconnect between systemic and mucosal responses. The consortium is in the process of measuring mucosal cytokines and we will look for an association with vitamin D when we analyse this dataset.

3.12 Page 18, line 17, consider inserting 'during illness' after '(insufficiency/deficiency; line 50 might need modification if allowing for the S shaped associations [see above] reveals anything useful).

We have made this change as suggested.

3.13 Page 19, line 53, you should mention that such increases in deficiency could well add to the problems of long-Covid here since that problem can cause diabetes as well as muscle weakness;

Our data for Covid-19 relate to the acute illness and unlike the non-selected ICU cohort we do not have measurements made on recovery in survivors. Long Covid likely has a complex pathophysiology and probably encompasses several distinct endotypes. We agree assessment of vitamin D status will be important in such cohorts (for example the PHOSP study) but we are reluctant to make speculative comments regarding the role of vitamin D in the current manuscript at a time when concrete data are needed.

3.14 Page 20, line 14+, there is very new data out from Barcelona in this area, using calcifediol treatment, so that this section could be updated?

Thank you for referring us to this data which we have now cited in our discussion of the results of vitamin D supplementation trials in Covid-19.

Reviewer 4 (Dr. D Papadimitriou)

4.1 Introduction L37: Here it could be added that in a recent ecological study [PMID: 34079693] it was shown that a higher 25(OH)D concentration may protect from serious-critical illness and death from COVID-19 disease - even more in the elderly - but does not seem to prevent severe acute respiratory syndrome coronavirus 2 from spreading.

Thank you for this additional reference. We had added it to the Introduction to support the statement re. geographic differences in prevalence of deficiency and Covid-19 mortality.

4.2 Introduction L41-42: This (with proper rephrasing) could be added to support the authors'

approach: measurement of binding proteins and free vitamin D metabolites may be essential

to create a more realistic approximation of vitamin D status [PMID: 23075939]

We have re-phrased the sentence re. the rationale of measuring free vitamin D and added the suggested reference.

4.3 A major limitation and important parameter to discuss wherever the authors judge as appropriate:

even if we had the possibility to measure at every infected person the total Vitamin D at the time of diagnosis, still it is possible that in those with an active disease, Vitamin D would have been lower than before infection, as it is rapidly “consumed” during illness [Caccialanza, R.; Laviano, A.; Lobascio, F.; Montagna, E.; Bruno, R.; Ludovisi, S.; Corsico, A.G.; Di Sabatino, A.; Belliato, M.; Calvi, M., et al. Early nutritional supplementation in non-critically ill patients hospitalized for the 2019 novel coronavirus disease (COVID-19): Rationale and feasibility of

a shared pragmatic protocol. Nutrition 2020, 74, 110835-110835, doi:10.1016/j.nut.2020.110835], indicating that after a high dose early short term supplementation to achieve levels of 25(OH)D3 > 100 nmol/L, a relatively high daily supplementation would be required during illness, before returning to safe long term supplementation that does not require medical supervision [PMID: 34079693].

We have included this in the revised discussion of the limitations of the study.

VERSION 2 – REVIEW

REVIEWER	Hewison, M University of Birmingham, Institute of Metabolism and Systems Research
REVIEW RETURNED	07-Sep-2021
GENERAL COMMENTS	This is an excellent paper but the modifications made by the authors have further improved the manuscript.
REVIEWER	Bychinin, Mikhail Federal Scientific and Clinical Center of Specialized Types of Medical Care and Medical Technologies of the Federal Medical and Biological Agency of Russia, Intensive Care Unit
REVIEW RETURNED	13-Sep-2021
GENERAL COMMENTS	A good revision that addresses the points raised previously.
REVIEWER	Boucher, Barbara Joan Queen Mary University of London, The Blizard Institute [Hon Professor]
REVIEW RETURNED	07-Sep-2021
GENERAL COMMENTS	Re bmjopen-2021-055435.R1

	This MS has been revised in line with the suggestions made by the various reviewers in so far as has been possible and I cannot see other changes that would be useful within the format of this particular study, bearing in mind that I am not a statistician and that a formal report on the statistical analyses should have been made, as I am sure it has been
--	--

REVIEWER	Papadimitriou, Dimitrios Athens Medical Centre, Pediatric-Adolescent Endocrinology and Diabetes
REVIEW RETURNED	26-Sep-2021

GENERAL COMMENTS	All reviewers' comments have been adequately addressed
--